# *Bothrops lanceolatus* snake venom impairs mitochondrial respiration and induces DNA release in human heart preparation

**Mariola Cano-Sanchez[1], Kais Ben-Hassen[2], Olivier Pierre Louis[1], Fabienne Dantin[3], Papa Gueye[4], Francois Roques[2], Hossein Mehdaoui[4], Dabor Resiere[4], Remi Neviere[1]***

**1** Cardiovascular Research Team EA7525, University of the French West Indies (Université des Antilles), Fort de France, France, **2** Department of Cardiovascular Surgery, CHU Martinique (University Hospital of Martinique), Fort-de-France, France, **3** Department of Biology, CHU Martinique (University Hospital of Martinique), Fort-de-France, France, **4** Department of Critical Care Medicine, Toxicology and Emergency, CHU Martinique (University Hospital of Martinique), Fort-de-France, France

\* remi.neviere@chu-martinique.fr

**Data Availability Statement:** All relevant data are within the manuscript and Supporting Information files.

## Abstract

### Introduction

Envenomations by *Bothrops* snakebites can induce overwhelming systemic inflammation ultimately leading to multiple organ system failure and death. Release of damage-associated molecular pattern molecules (DAMPs), in particular of mitochondrial origin, has been implicated in the pathophysiology of the deregulated innate immune response.

### Objective

To test whether whole *Bothrops lanceolatus* venom would induce mitochondrial dysfunction and DAMPs release in human heart preparations.

### Methods

Human atrial trabeculae were obtained during cannulation for cardiopulmonary bypass from patients who were undergoing routine coronary artery bypass surgery. Cardiac fibers were incubated with vehicle and whole *Bothrops lanceolatus* venom for 24hr before high-resolution respirometry, mitochondrial membrane permeability evaluation and quantification of mitochondrial DNA.

### Results

Compared with vehicle, incubation of human cardiac muscle with whole *Bothrops lanceolatus* venom for 24hr impaired respiratory control ratio and mitochondrial membrane permeability. Levels of mitochondrial DNA increased in the medium of cardiac cell preparation incubated with venom of *Bothrops lanceolatus*.

### Conclusion

Our study suggests that whole venom of *Bothrops lanceolatus* impairs mitochondrial oxidative phosphorylation capacity and increases mitochondrial membrane permeability. Cardiac

**Funding:** This work was supported by the Agence Nationale de la Recherche (ANR-18-CE17-0026-01 to RN; ANR-18-CE17-0026-03 to DR) https://anr.fr/Project-ANR-18-CE17-0026. The funders had no role in study design, data collection and analysis, decision to publish, or preparation of the manuscript.

**Competing interests:** The authors have declared that no competing interests exist.

mitochondrial dysfunction associated with mitochondrial DAMPs release may alter myocardium function and engage the innate immune response, which may both participate to the cardiotoxicity occurring in patients with severe envenomation.

## Author summary

Despite initial symptomatic management and adequate antivenin strategy, highly venomous Bothrops snakebites frequently induce overwhelming inflammation leading to multiple organ system failure and death. We state that recognition of venom-associated molecular patterns and cellular damage-associated molecular pattern molecules (DAMPs) by pattern-recognition receptors will engage inflammation and cell-mediated immune response. Due to endosymbiotic bacterial origin of mitochondria, mitochondrial DAMPs released from injured envenomed tissues are recognized as danger signals and exacerbate the innate inflammatory host response. Hence, mitochondrial DAMPs will engage a vicious circle, which deregulates inflammation via aberrant mitochondrial signaling, impaired mitophagy and disruption of mitochondrial dynamics. Delineating critical factors that elicit mtDAMPs release will generate hypothesis for new treatments.

## Introduction

Envenomation induced by *Bothrops* snake bites presents a vast spectrum of clinical severity ranging from mild cases, with local edema and pain as predominant manifestations, to severe cases, in which a series of life-threatening manifestations may occur [1]. Bleeding, coagulopathy, edema, and hemodynamic alterations are typical clinical manifestations of *Bothrops* snake envenomation [1,2]. Mechanisms of these devastating local and systemic manifestations are attributed to the direct effects of toxins contained in snake venom predominately including snake venom metalloproteinases, snake venom serine proteinases, phospholipases $A_2$, C-type lectin-like toxins, disintegrins, cysteine-rich secretory proteins, and L-amino acid oxidases [1].

Toxins delivered to the victims can also trigger a potent systemic inflammatory response host defense accompanied by the release of a multitude of mediators and signaling molecules leading to edema, endothelial activation, leukocyte migration and prothrombotic features [3,4]. Stimulation of the innate immune response in snakebite envenomation has been attributed to the expression of surface and cytosolic Toll-Like Receptors (TLRs) enabling the recognition of molecular patterns such damage and venom associated molecular patterns, i.e., DAMPs and VAMPS [5–8]. Once engaged, TLRs trigger signaling pathways that culminate in the transcription of inflammatory genes boosting the inflammatory response systems. These events can progress to either resolution or an excessive and uncontrolled systemic inflammatory response leading to multiple organ failure and death [1,2].

Within the last decade, mounting evidence suggest that damaged mitochondria activate innate immune pathways [9]. Because of their bacterial origin, parallels between how cells respond to mitochondrial and bacterial ligands are not altogether surprising. Mitochondrion-driven innate immunity has been related to the ability of mitochondrial DAMPs release (e.g., mtDNA, cardiolipin, formyl-methionine-labeled peptides, and cytochrome c) to engage pattern recognition receptors such as TLRs and trigger the inflammatory cascade [9,10]. Although the importance of mitochondrial DAMPs in triggering inflammation is well-recognized in many human pathologies, studies on their participation in snake venom-induced

inflammation are scarce [7,8]. Previous experiments have demonstrated that *Bothrops* snake venom elicits mitochondrial DNA (mtDNA) and cytochrome c release in *ex vivo* treatment of tibialis anterior muscles [7]. In addition, injection of *Bothrops* snake venom in mice induced the release of mtDNA and cytochrome c in the circulation [7]. Overall, experimental evidence suggest that mitochondrial DAMPs release would mediate inflammatory signals in the envenomed tissue environment. Such mechanism has not been demonstrated in humans. As mitochondria account for $\sim 35\%$ of the cardiomyocyte volume, human heart preparations can provide a suitable model for the evaluation of the impact of snake venoms on mitochondrial function.

The aim of the present study was to test whether whole *Bothrops lanceolatus* venom would induce mitochondrial dysfunction and DAMPs release in human heart preparations. Specifically, we assessed mitochondrial respiration in human cardiac permeabilized fibers and the release of mtDNA of human myocardium preparation exposed to venom mixture.

## Materials and methods

### Ethics statement

The study was conducted in accordance with the amended Declaration of Helsinki (http://www.wma.net/en/30publications/10policies/b3/). All patients gave their informed consent for the processing of personal data and tissue collection for scientific research purposes. The study was approved by the French medical ethics committee (CPP: 19-MART-01).

### Human right atrial trabeculae

Right atrial appendages were obtained during cannulation for cardiopulmonary bypass from patients who were undergoing routine coronary artery bypass surgery. All patients were anaesthetized with target-controlled administration of sevoflurane or propofol, sufentanil, and skeletal muscle relaxant. Patients with atrial arrhythmia or history of atrial arrhythmia and those with left ventricular ejection fraction $< 55\%$ were excluded from the study. Atrial appendages obtained during surgery were placed in ice-cold Krebs-Henseleit buffer (KHB), and immediately delivered to laboratory.

### Human atrial myocardium preparations

Atrial appendages were carefully prepared under stereomicroscopic control in ice-cold BIOPS relaxing solution as we have previously described [11]. Fibers were separated with sharp forceps and transferred to vials containing 2 ml of oxygenated (95% $O_2$, 5% $CO_2$) BIOPS buffer containing 2.77 mM $CaK_2EGTA$, 7.23 mM $K_2EGTA$, 6.56 mM $MgCl_2$, 0.5 mM dithiothreitol, 50 mM K-MES, 20 mM imidazole, 20 mM taurine, 5.3 mM $Na_2ATP$, 15 mM phosphocreatine, pH 7.1. at 37°C. Whole *Bothrops lanceolatus* venom (10 and 100 μg/ml) (Latoxan Laboratory, Portes lès Valence, France) or vehicle, i.e., human serum albumin in equivalent volume (10 and 100 μg/ml) (Sigma Aldrich, France) were added to the vials. After 24hr of incubation, fibers and supernatants were separated by centrifugation (50g at 4°C for 10 minutes) and used for measurements of mitochondrial respiration and mtDNA quantification, respectively.

### High-resolution respirometry

At the end of incubation time, fibers (10 mg) were transferred into 2mL of fresh ice-cold BIOPS relaxing solution and 20μL of saponin (freshly-prepared stock solution 5mg/mL) are added to obtain final concentration of 50μM. Bundles were maintained for 30 min with saponin solution, and then washed for 5 min three times in ice-cold respiration medium MiRO5

containing 110 mM sucrose, 20 mM HEPES, 10 mM $KH_2PO_4$, 20 mM taurine, 3 mM $MgCl_2$ $6H_2O$, 60 mM MES-K, 0.5 mM EGTA and 0.1% bovine serum albumin; pH 7.1. Respiration rates were measured ($O_2$k oxygraph, Oroboros, Innsbruck, Austria) and expressed in pico-moles $O_2$ per second per milligram wet weight. Data acquisition and analysis were performed with Datlab4 software (Oroboros, Innsbruck, Austria).

Substrate and/or inhibitors for respiratory experiments were added within $MiRO_5$ in a step-by-step manner using micro syringes. Chemical agents (Sigma Aldrich, France) were sequentially prepared according to Oroboros manufacturer data sheet and administered as we have previously described [12]. Respiration rates were measured ($O_2$k oxygraph, Oroboros, Innsbruck, Austria) and expressed in picomoles $O_2$ per second per milligram wet weight.

Chemical agents (Sigma Aldrich, France) were sequentially prepared according to Oro-boros manufacturer data sheet and administered as described below:

- Complex I-dependent state 2 respiration ($V_0$) was determined as the respiration rate in the presence of 10 mM L-glutamate and 2 mM L-malate dissolved in $H_2O$, which activate the Krebs cycle enzyme malate dehydrogenase providing NADH to complex I of the respiratory chain.

- Complex I-dependent state 3 respiration ($V_{glut+mal}$) was determined as a phosphorylation-stimulated respiration rate in the presence of 5 mM ADP added to the $V_0$ medium providing ADP to $F_1$-$F_0$ ATP synthase. The coupling of phosphorylation to oxidation was estimated by calculating the respiratory control ratio (RCR) as the ratio $V_{glut+mal}$ / $V_0$.

- Intactness of mitochondrial outer membrane was evaluated by the mean of cytochrome c. In brief, respiration in the presence of 10μM cytochrome c dissolved in ethanol ($V_{cyt-c}$) (horse heart cytochrome c, Sigma Aldrich, France) was determined. $V_{cyt-c}/V_{glut+mal}$ ratio was used as a permeability index of the outer mitochondrial membrane.

- Complex I+II-dependent state 3 respiration ($V_{succ+glut}$) was determined as a phosphoryla-tion-stimulated respiration rate in the presence of 10 mM succinate dissolved in $H_2O$ and added to the $V_{cyt-c}$ medium activating the Krebs cycle enzyme succinate dehydrogenase pro-viding $FADH_2$ and the complex II (succinate dehydrogenease) of the respiratory chain.

- Complex I+II-dependent respiration ($V_{succ+glut}$) was then uncoupled by addition of 10 μM carbonyl cyanide p-(trifluoro-methoxy) phenyl hydrazone (FCCP), an oxidative phosphory-lation uncoupler, dissolved in ethanol and added to the $V_{succ+glut}$ medium. FCCP makes the inner mitochondrial membrane permeable to protons and fully activates mitochondrial res-piration. Dissipation of the proton gradient along the inner mitochondrial membrane inhib-its ADP phosphorylation by $F_1$-$F_0$ ATP synthase.

- Complex II-dependent uncoupled state of respiration ($V_{rot}$) was determined as the respira-tion rate by adding 0.5 μM of complex I inhibitor rotenone dissolved in ethanol to the medium.

- $O_2$ consumption independent of the respiratory-chain ($V_{AA}$) was determined as the respira-tion rate in presence of complex III inhibitor antimycin-A dissolved in ethanol (2.5 μM) and added to the $V_{rot}$ medium. Thus, electron supply to respiratory chain complex IV, and respi-ratory chain respiration were stopped.

- Complex IV-dependent uncoupled state of respiration ($V_{COx}$) was determined in two steps. $O_2$ consumption was first determined in presence of 2 mM ascorbate and 0.5 mM N,N,N', N'-tetramethyl-p-phenylenediamine dihydrochloride TMPD dissolved in $H_2O$ and added to the $V_{AA}$ medium ($V_{TMPD-asc}$). TMPD was used as an artificial redox mediator that assists

electron transfer from ascorbate to cytochrome c. TMPD-ascorbate auto-oxidation was then determined as the $O_2$ utilization ($O_2$ consumption) in presence of 1 mM KCN added to the $V_{TMPD-asc}$ medium. Complex IV-dependent uncoupled state of respiration ($V_{COx}$) was calculated as $V_{TMPD-asc}$ minus $V_{KCN}$. As antimycin-A inhibits complex III, $V_{COx}$ estimates complex IV-related maximum respiration rate.

• At the end of each experiment, chambers were calibrated for zero O2 content with dithionite.

## Mitochondrial membrane integrity

Mitochondrial membrane integrity was indirectly evaluated by measuring reduction of exogenous cytochrome c (Cyt c) [13]. In healthy mitochondria, Cyt c is located in the mitochondrial intermembrane/inter-cristae spaces, where it functions as an electron shuttle to drive the respiratory chain activity. Impaired mitochondrial membrane permeability allows more cytochrome c to enter the mitochondria and stimulates mitochondrial respiration.

## Mitochondrial DNA (mtDNA) copy number and real-time PCR

At the end of incubation time, supernatant (200 μL) were treated with RNAse A (100 mg/ml) to avoid RNA contamination and mtDNA was extracted using QIAamp DNA Mini Kit (Qiagen, France) following manufacturer's instructions. The relative mtDNA copy number was measured by PCR and corrected by simultaneous measurement of the nuclear DNA. The forward and reverse primers for mtDNA which are complementary to the sequence of the human mitochondrial ND2 gene were ND2 forward primer (*TAAAACTAGGAATAGCCCCC*) and reverse primer (*TTGAGTAGTAGGAATGCGGT*) [14] and sequences complementary to the 18S gene were the primers used for the detection of nuclear DNA. Quantitative PCR was performed on an Eppendorf Realplex S2 (Eppendorf, Germany) using Mesa Blue Mix (Eurogentec, France). Mitochondrial DNA levels were adjusted for nuclear DNA levels and analyzed using the $^{\Delta\Delta Ct}$ method.

## Statistics

Numerical data are given as mean ± standard deviation (SD). The Shapiro-Wilk test was used to test for normal distribution of numerical data. Data were analyzed by using Student-t test and analysis of variance ANOVA. When a significant difference was found, we identified specific differences between groups with a sequentially rejective Bonferroni procedure. After application of the Bonferroni correction, $p < 0.05$ was taken as a level of statistical significance. Results were analyzed with the SPSS for Windows software, version 24.0 (SPSS Paris-la-Defense, France).

## Results

We assessed functional properties of mitochondrial respiration of human heart preparations in dependence on the storage time as previously described. Mitochondrial respiration was evaluated in cardiac fibers immediately permeabilized after a 24hr storage period as described in Materials and methods. Long temporal stability of mitochondrial function in human cardiac muscle fibers is summarized Table 1. Pilot experiments demonstrated that compared with vehicle, incubation of cardiac fibers with whole *Bothrops lanceolatus* venom for 3hr and 6hr had no effects on mitochondrial respiration (n = 2 in each experiments). Compared with

**Table 1. Respiration rates of control cardiac human fibers in dependence on the storage time.**

| | glutamate malate | | Cyt c | succinate | rotenone | TMPD/asc |
|---|---|---|---|---|---|---|
| | $V_0$ | $V_{glut+mal}$ | $V_{cyt-c}/V_{glut+mal}$ | $V_{succ}$ | $V_{rot}$ | $V_{COx}$ |
| T0 | 3.1±2.3 | 42.2±6.4 | 48.1±3.4 | 48.6±2.1 | 32.0±7.1 | 72.9±11.9 |
| T24h | 4.3±0.8 | 41.3±8.5 | 45.5±5.5 | 46.7±2.5 | 30.4±6.5 | 68.0±9.3 |
| p-value | 0.255 | 0.840 | 0.348 | 0.185 | 0.693 | 0.445 |

Rates of respiration are given in picomoles $O_2$ per second per milligram wet weight (pmol/$O_2$/sec/mg) at baseline and following 24hr storage in BIOPS medium. See Materials and Methods for detailed description. $V_0$ complex I-dependent state 2 respiration; $V_{glut+mal}$ complex I-dependent state 3 respiration; $V_{succ}$ complex I+II-dependent state 3 respiration; $V_{rot}$ complex II-dependent uncoupled state of respiration; $V_{Cox}$ complex IV-dependent uncoupled state of respiration. Data are mean ±SEM. Results were analyzed with Student t-tests (n = 6 in each group).

vehicle, incubation of human cardiac muscle with whole *Bothrops lanceolatus* venom for 24hr consistently induced respiratory control ratio reduction (Fig 1A). Respiration rates expressed in picomoles $O_2$ per second per milligram wet weight are summarized Table 2 and S1 Data.

Exogenous Cyt c induced mitochondrial respiration increased less than 5% (Fig 1B) in human cardiac muscle incubated with vehicle for 24hr. In contrast, exogenous Cyt c induced significant increases of mitochondrial respiration in human cardiac muscle incubated with whole *Bothrops lanceolatus* venom for 24hr, suggesting impairment of mitochondrial membrane permeability.

In order to extend the analysis of mitochondrial permeability, quantification of mtDNA in the supernatant of cardiac cell preparation incubated with vehicle and whole *Bothrops lanceolatus* venom for 24hr were performed. Fig 1C shows that mtDNA was detectable in the medium following incubation of cardiac cell preparation with whole *Bothrops lanceolatus* venom for 24hr (S2 Data).

## Discussion

Snake envenomation is a common but neglected disease that affects millions of people around the world annually [1]. In the overseas French territories, snakebites are related to the presence of endemic venomous snake species such as *Bothrops atrox*, *Bothrops brazili*, and *Lachesis muta* in French overseas territories and *Bothrops lanceolatus* in Martinique [15,16]. In these overseas territories, *Bothrops* snakebites are responsible for life-threatening envenomations related to an overwhelming systemic inflammatory and hemostatic host response. Proposed mechanisms of these complications include the release of damage-associated molecular pattern molecules (DAMPs), in particular of mitochondrial origin, which triggers the systemic inflammatory response [5–8]. Here for the first time, we have shown in human heart preparation that venom of *Bothrops lanceolatus* causes mitochondrial dysfunction and mitochondrial DNA release, which is regarded as an endogenous danger signal inducing the inflammatory response.

Firstly, we found that venom of *Bothrops lanceolatus* caused reduction of respiratory control ratio in human heart preparation, suggesting that efficiency of mitochondrial oxidative phosphorylation was impaired. Previous studies in human cell lines have shown that toxins isolated from *Bothrops* snake venom can promote mitochondrial dysfunction [17–19]. Changes of mitochondrial respiration induced by snake venoms in human tissue have not been previously reported. Here, we have shown that venom of *Bothrops lanceolatus* deteriorates oxidative phosphorylation capacity in human cardiac preparation. Further experiments using contracting isolated human right atrial trabeculae are warranted to demonstrate that

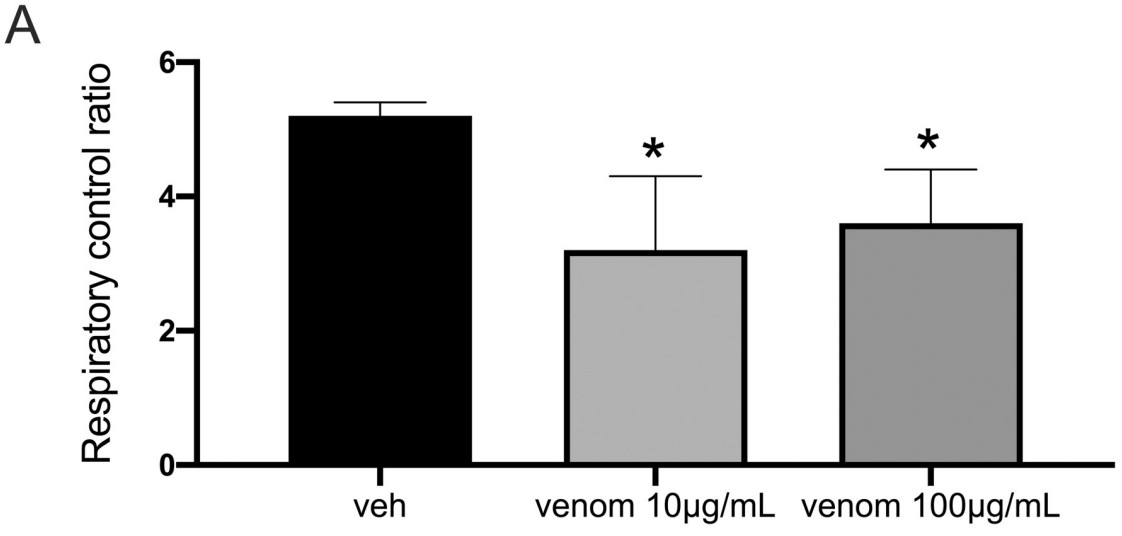

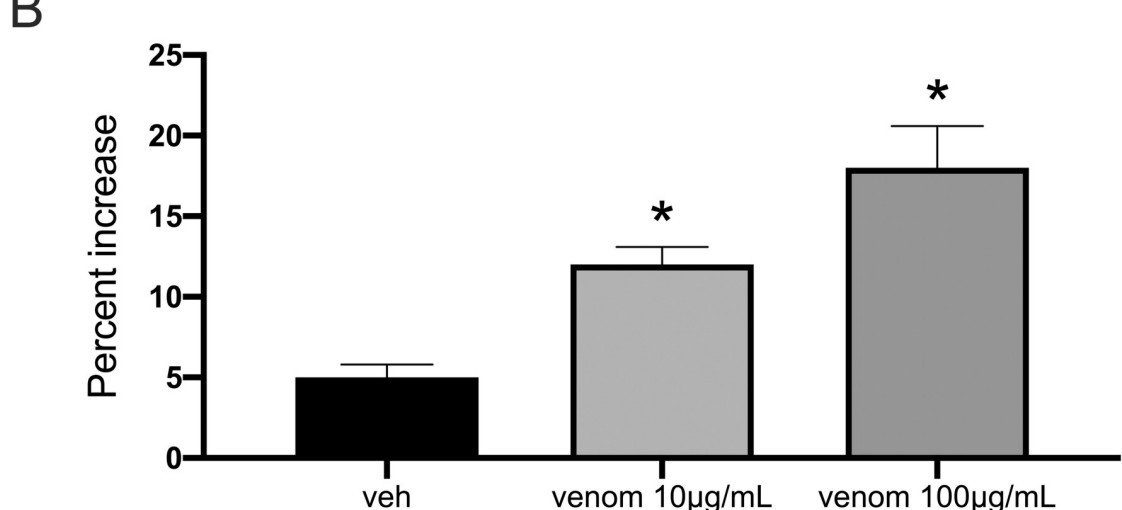

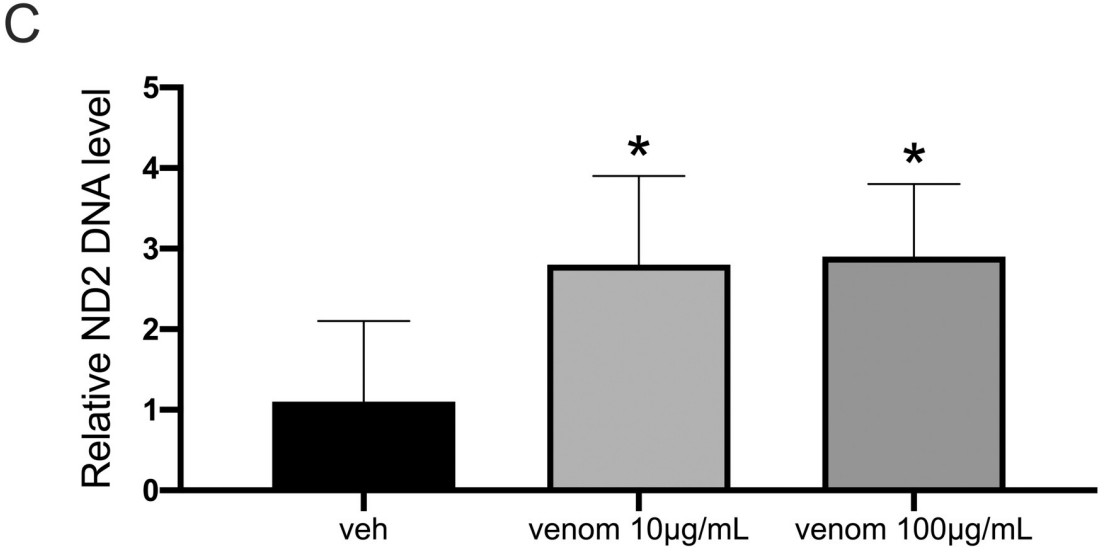

**Fig 1.** Respiratory control ratio (**A**), effects of exogenous Cyt c (**B**) on respiratory rate expressed as $(V_{cytc}—V_{glut+mal})/V_{glut+mal}$ and medium release of mtDNA (**C**) in cardiac preparations treated with vehicle and whole *Bothrops lanceolatus* venom for 24hr. Relative ND2 DNA levels to 18S nuclear gene expression. Data are mean ± SD (n = 8). * indicates p<0.05.

such oxidative phosphorylation deficits may be associated with cardiac contractile dysfunction, which can occur in pitviper snake envenoming [20,21].

Secondly, we found that venom of *Bothrops lanceolatus* impaired mitochondrial membrane permeability, which was evaluated indirectly by measuring reduction of exogenous cytochrome c (Cyt c) in the medium of cardiac fiber preparation. Cyt c, a peripheral protein, only loosely bound of the mitochondrial inner membrane, functions as an electron shuttle between complex III and complex IV of the respiratory chain [13]. If the outer membrane of mitochondria is damaged, the endogenous Cyt c can be released from intermembrane space at physiological ionic strength and will inhibit respiration. In high-resolution respirometry studies, Cyt c is applied to test the integrity of the mitochondrial outer membrane [13]. Herein, when outer membrane of mitochondria is intact the addition of exogenous Cyt c has no effect on respiration. In contrast, addition of exogenous Cyt c to the experimental preparation will markedly stimulate the respiratory rate when the outer membrane of mitochondria has been damaged. In our study, addition of exogenous Cyt c induced significant increases of mitochondrial respiration in human cardiac muscle incubated with whole *Bothrops lanceolatus* venom, suggesting impairment of mitochondrial membrane permeability. In order to extend the evaluation of mitochondrial permeability increase, we evaluated the release of mtDNA in the medium of cardiac cell preparation incubated with venom of *Bothrops lanceolatus*. Consistently with previous experiments showing that mitochondrial DNA, cytochrome c, and ATP are released in tissue affected by venoms of *Bothrops* [7,8], we confirm that whole *Bothrops lanceolatus* venom can induce the release of mitochondrial DNA in human heart tissue.

Overall, it is likely that mitochondrial DNA, a ligand of several innate immune receptors, would be involved in intensity of the inflammatory response in envenomed patients. Indeed, mitochondria are cellular organelles that orchestrate several biological processes, ranging from energy production and metabolism to cell death and inflammation. Release of mitochondrial DNA into the cytoplasm and out into the extracellular milieu activates a plethora of different pattern recognition receptors and innate immune responses, including TLRs and inflammasome formation leading to, among others, robust type I interferon responses [22]. Upon stress or cellular damage, one group of DAMPs, the mitochondrial nucleic acids, are released from their compartments and sensed as foreign, eliciting a similar innate immune response one would see against pathogens. Most of the sensors of mtDNA have only been recently identified [22]. Amongst all the cytosolic DNA sensors, cyclic GMP–AMP synthase (cGAS), activating

**Table 2. Effects on respiration rates of 24hr incubation of cardiac human permeabilized fibers with vehicle and whole *Bothrops lanceolatus* venom.**

|  | glutamate malate | | succinate | rotenone | antimycin-A | TMPD/asc |
|---|---|---|---|---|---|---|
|  | $V_0$ | $V_{glut+mal}$ | $V_{succ + glut}$ | $V_{rot}$ | $V_{AA}$ | $V_{COx}$ |
| vehicle | 4.3±0.8 | 41.3±8.5 | 46.7±2.5 | 30.4±6.5 | 3.3±0.5 | 68.0±9.3 |
| 10μg/ml | 3.7±1.3 | 17.3±1.4 * | 16.7±2.5* | 10.4±2.5 * | 1.8±1.3 | 62.0±2.9 |
| 100μg/ml | 4.1±0.9 | 10±0.8 * | 12.5±0.3 * | 8.5±0.6 * | 2.1±1.8 | 38.0±7.1 * |

Rates of respiration are given in picomoles $O_2$ per second per milligram wet weight (pmol/$O_2$/sec/mg) in vehicle and venom-treated cardiac preparation. See Materials and Methods for detailed description. $V_0$ complex I-dependent state 2 respiration; $V_{glut+mal}$ complex I-dependent state 3 respiration; $V_{succ}$ complex I+II-dependent state 3 respiration; $V_{rot}$ complex II-dependent uncoupled state of respiration; $V_{Cox}$ complex IV-dependent uncoupled state of respiration. Data are mean ± SD (n = 8). * indicates p<0.05.

endoplasmic reticulum-resident Stimulator of interferon genes (STING) are probably the most explored pathway. This pathway is triggered during infection with cytosolic bacterial pathogens and some DNA viruses resulting into transcriptional induction of type I interferon and the nuclear factor-κB (dependent expression of proinflammatory cytokines [23].

## Study limitations

This study had several limitations. First, the study mainly documented the effects of incubation with whole *Bothrops lanceolatus* venom after 24hr, whereas shorter time intervals that would induce similar mitochondrial alterations were not studied in details. Identification of DAMPs released in the supernatant of cardiac cell preparation incubated with vehicle and whole *Bothrops lanceolatus* venom was limited to mtDNA. Besides, it was not possible to identify the toxins responsible for the mitochondrial effects as experiments were done with crude venom. Eventually, whether changes in oxidative phosphorylation induced by whole *Bothrops lanceolatus* venom can induce cardiac contractile dysfunction was not studied.

## Conclusion

Our study suggests that incubation of human cardiac cell preparation with whole *Bothrops lanceolatus* venom for 24hr impairs mitochondrial oxidative phosphorylation capacity and elicits the release of mitochondrial DNA, a known trigger of the innate immune response.

## Supporting information

**S1 Data. Supporting information file on mitochondrial respiration protocols and results.**
(XLSX)

**S2 Data. Supporting information file on RT PCR raw data for human mitochondrial ND2 gene**
(XLSX)

## Author Contributions

**Conceptualization:** Kais Ben-Hassen, Dabor Resiere, Remi Neviere.

**Formal analysis:** Mariola Cano-Sanchez, Olivier Pierre Louis, Fabienne Dantin, Hossein Mehdaoui, Dabor Resiere, Remi Neviere.

**Investigation:** Mariola Cano-Sanchez, Kais Ben-Hassen, Olivier Pierre Louis, Hossein Mehdaoui, Dabor Resiere, Remi Neviere.

**Methodology:** Mariola Cano-Sanchez, Kais Ben-Hassen, Dabor Resiere, Remi Neviere.

**Supervision:** Fabienne Dantin, Papa Gueye, Francois Roques.

**Writing – original draft:** Dabor Resiere, Remi Neviere.

**Writing – review & editing:** Remi Neviere.

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
