## [Decision Letter · Decision Letter 0]

16 Apr 2022

Dear Pr NEVIERE,

Thank you very much for submitting your manuscript "Bothrops lanceolatus snake venom impairs mitochondrial respiration and induces DNA release in human heart preparation" for consideration at PLOS Neglected Tropical Diseases. As with all papers reviewed by the journal, your manuscript was reviewed by members of the editorial board and by several independent reviewers. In light of the reviews (below this email), we would like to invite the resubmission of a significantly-revised version that takes into account the reviewers' comments. 

We cannot make any decision about publication until we have seen the revised manuscript and your response to the reviewers' comments. Your revised manuscript is also likely to be sent to reviewers for further evaluation.

Sincerely,

Kartik Sunagar, Ph.D.

Deputy Editor

Kartik Sunagar

Deputy Editor

Reviewer's Responses to Questions

**Key Review Criteria Required for Acceptance?**

**Methods**

-Are the objectives of the study clearly articulated with a clear testable hypothesis stated?

-Is the study design appropriate to address the stated objectives?

-Is the population clearly described and appropriate for the hypothesis being tested?

-Is the sample size sufficient to ensure adequate power to address the hypothesis being tested?

-Were correct statistical analysis used to support conclusions?

-Are there concerns about ethical or regulatory requirements being met?

Reviewer #1: The objectives of this study are clear and well presented. The study design is appropriate and the population is clearly described, as well as the methods used. Appropriate statistical analyses were carried out. I have no ethical concerns with the study.

Line 101: The term ‘purified venom’ is a bit misleading since the term ‘purified’ usually refers to the use of isolated toxins. In this case the authors are using crude or whole venom. I suggest using the term ‘whole B. lanceolatus venom' throughout the paper instead of ‘purified venom’.

Line 103: The authors only used one incubation time (24 hr). It would have been appropriate to use shorter incubation times as well, for example 3 hr or 6 hr, to have a view on the early effects of venom in the system. That would provide a time-course of the effects being studied.

Reviewer #2: Please refer general comments

**Results**

-Does the analysis presented match the analysis plan?

-Are the results clearly and completely presented?

-Are the figures (Tables, Images) of sufficient quality for clarity?

Reviewer #1: The analysis presented matched the analysis plan and are clearly formulated. Results are clear and well presented. The figures and tables are also clear and summarize the main findings of the study.

Line 174: The text is confusing with the use of the word ‘either’. Please revise.

Reviewer #2: Please refer general comments

**Conclusions**

-Are the conclusions supported by the data presented?

-Are the limitations of analysis clearly described?

-Do the authors discuss how these data can be helpful to advance our understanding of the topic under study?

-Is public health relevance addressed?

Reviewer #1: Most of the conclusions are supported by the data, with one exception: Line 226: The extrapolation of these observations to the clinical situation should be made with caution. In the experimental setting used. Cardiac tissue is directly exposed to venom for a prolonged period of time (24 hr). This is very different from a real envenoming case since (a) previous toxicokinetic data indicate that venom concentration in cardiac tissue after experimental envenoming in vivo is very low, and (b) the vasculature of the heart is of the ‘continuous’ type, meaning that it has low permeability and, therefore, the amount of venom that would reach myocardial cells is probably low in vivo. Thus, even though the observations clearly demonstrate that this venom is able to affect cardiac tissue and release DAMPs, the extrapolation of these findings to explain cardiac effects in snakebites is too speculative. This comment also applies to the conclusion paragraph (lines 248-250).

The authors present a well structured discussion on how these findings help to explain one important aspect of the action of snake venoms in tissues, i.e., the release of DAMPs which might impact on the innate immune response, with a possible effect on the systemic outcome of envenomings.

The authors did not discuss the limitations of their work, some of which are: (1) the incubation time is 24 hr and it is not clear whether shorter time intervals would induce similar alterations. (2) The work was done with crude venom, and this does not allow the identification of the toxins responsible for the effects. 

Line 212: It should be Bothrops atrox, B. brazili and Lachesis muta

Line 213: It should be ‘French overseas territories’

Reviewer #2: Please refer general comments

**Editorial and Data Presentation Modifications?**

Reviewer #1: (No Response)

Reviewer #2: Please refer general comments

**Summary and General Comments**

Reviewer #1: This study presents an interesting contribution that expands our understanding on the action of the venom of B. lanceolatus. In recent years, the concept that snake venoms induce tissue damage with the release od DAMPs, and that these molecules may play a role in the overall pathophysiology of envenoming, contributing to systemic inflammation, has gained support. This study demonstrates with a sound experimental setting that the venom of B. lanceolatus is able to impair mitocondrial respiratory funcion in human cardiac tissue and also to release DAMPs from the tissue. 

My main concern has to do with the extrapolation made by the authors on the possible implications of their findings for the clinical cardiotoxicity induced by this venom. In my comments to the Discussion I have mentioned why I believe this extrapolation is too speculative and not necessarily suported by their findings. The main conclusion of this study should be focused on the ability of venom to impair mitochondrial function and to release DAMPS from the tissue, without extrapolating this to explain cardiac dysfunction in envenomings. For this, the authors should use in vivo experimental systems.

Reviewer #2: The study on Bothrops lanceolatus snake venom impairs mitochondrial respiration and induces DNA release in human heart preparation (PNTD-D-22-00263) by Cano-Sanchez et al have tried to address the effect of the venom on the cardiac muscle fibres, especially on the role of mitochondria in the venom induced toxicity. The authors have made an interesting observation of leaking of mitochondrial DNA which they consider it as the sole DAMP involved in venom induced cardiac related systemic toxicity. It is interesting that the authors have also tried to connect the mitochondrial DNA release to the innate immune response. The manuscript may be accepted after addressing the following queries.

Minor comments:

Line 65 – remove receptor from ‘Toll-Like Receptor (TLR) receptors’.

Line 103 - what is the vehicle control used?

Line 135- mention the concentration of antimycin-A.

Line 192 - In table 2, third column doesn’t match the description of the legend.

Line 195, 197,198 - There is no uniformity in manuscript writing, for example, VAA is written as VAA.

Line 196 - Correct the word ‘Vsucc+glu’.

Line 106 – methodology for, ‘High-resolution respirometry’ is clumsy and difficult to follow. Split the methodology and explain in detail.

Major comments:

What is the basis for Bothrops lanceolatus venom targeting cardiac muscle fibres only?

The authors have to discuss about the fate of mitochondria after releasing their DNA. It is even better if they demonstrate the extent of their viability.

Inclusion of the data on nuclear DNA release will make the readership better. 

As the study focuses on the mitochondrial dysfunction, it is essential to shed light on the possible role of mitochondrial membrane lipids, example lipid peroxidation, cardiolipin role etc if any.

The authors have focussed only on the release of mitochondrial DNA, it is important that they should address how and what promote the DNA release. Is the venom/venom toxin(s) exerting the DNA release effect directly interacting with the mitochondria (after crossing the sarcolemma) or does it acts at the sarcolemma level? An insight in to the mechanism of action will provide better clarity.

It is important for the authors to demonstrate the state of respiration of mitochondria in venom treated cardiac fibres without saponin treatment.

Mitochondrial DNA release and its connectivity to innate immunity need to be discussed properly.

PLOS authors have the option to publish the peer review history of their article (what does this mean?). If published, this will include your full peer review and any attached files.

Reviewer #1: No

Reviewer #2: No
---

## [Decision Letter · Decision Letter 1]

20 May 2022

Dear Pr NEVIERE,

We are pleased to inform you that your manuscript 'Bothrops lanceolatus  snake venom impairs mitochondrial respiration and induces DNA release in human heart preparation' has been provisionally accepted for publication in PLOS Neglected Tropical Diseases.

Best regards,

Kartik Sunagar, Ph.D.

Deputy Editor

Kartik Sunagar

Deputy Editor

Reviewer's Responses to Questions

**Key Review Criteria Required for Acceptance?**

**Methods**

-Are the objectives of the study clearly articulated with a clear testable hypothesis stated?

-Is the study design appropriate to address the stated objectives?

-Is the population clearly described and appropriate for the hypothesis being tested?

-Is the sample size sufficient to ensure adequate power to address the hypothesis being tested?

-Were correct statistical analysis used to support conclusions?

-Are there concerns about ethical or regulatory requirements being met?

Reviewer #1: See general comments

Reviewer #2: Refer general comments

**Results**

-Does the analysis presented match the analysis plan?

-Are the results clearly and completely presented?

-Are the figures (Tables, Images) of sufficient quality for clarity?

Reviewer #1: See general comments

Reviewer #2: Refer general comments

**Conclusions**

-Are the conclusions supported by the data presented?

-Are the limitations of analysis clearly described?

-Do the authors discuss how these data can be helpful to advance our understanding of the topic under study?

-Is public health relevance addressed?

Reviewer #1: See general comments

Reviewer #2: Refer general comments

**Editorial and Data Presentation Modifications?**

Reviewer #1: (No Response)

Reviewer #2: Refer general comments

**Summary and General Comments**

Reviewer #1: The authhors have adequately aqddressed my concerns and suggestions to the first version of this manuscript. The modifications introduced are satisfactory.

Reviewer #2: The authors have studied the effect of Bothrops lanceolatus venom on the mitochondrial dysfunction and DAMPs release in human heart preparations. The observed effects were demonstrated in permeabilized cardiac fibers.

The main concern was that the authors have demonstrated the mitochondrial dysfunction and DAMPs release in permeabilized cardiac fibers. Therefore, they were asked to demonstrate the effects in non-permeabilized cardiac fibers. That is treating the cardiac fibers directly with the venom, and also the signaling pathway. These have not been addressed.

In this study, the role of TLRs, inflammasome formation, and type-I interferon responses are all speculative. Therefore need justifications.

The last, but one sentence in the abstract is too speculative. ‘Hence, mitochondrial DAMPs will engage a vicious circle, which deregulates inflammation via aberrant mitochondrial signaling, impaired mitophagy, and disruption of mitochondrial dynamics. In this study, no evidence has been provided to justify the following statement.

Mitochondrial dysfunction and release of mitochondrial DNA (DAMP) are good observations. Therefore, this manuscript may be accepted as a short report.

PLOS authors have the option to publish the peer review history of their article (what does this mean?). If published, this will include your full peer review and any attached files.

Reviewer #1: No

Reviewer #2: No

---

## [Editor Report · Acceptance letter]

16 Jun 2022

Dear Pr NEVIERE,

We are delighted to inform you that your manuscript, "Bothrops lanceolatus  snake venom impairs mitochondrial respiration and induces DNA release in human heart preparation," has been formally accepted for publication in PLOS Neglected Tropical Diseases.

Best regards,

Shaden Kamhawi

co-Editor-in-Chief

Paul Brindley

co-Editor-in-Chief
